# Ways to Improve Insights into Clindamycin Pharmacology and Pharmacokinetics Tailored to Practice

**DOI:** 10.3390/antibiotics11050701

**Published:** 2022-05-21

**Authors:** Laura Armengol Álvarez, Greet Van de Sijpe, Stefanie Desmet, Willem-Jan Metsemakers, Isabel Spriet, Karel Allegaert, Jef Rozenski

**Affiliations:** 1Rega Institute for Medical Research, KU Leuven, B-3000 Leuven, Belgium; jef.rozenski@kuleuven.be; 2Department of Pharmaceutical and Pharmacological Sciences, KU Leuven, B-3000 Leuven, Belgium; greet.vandesijpe@kuleuven.be (G.V.d.S.); isabel.spriet@uzleuven.be (I.S.); karel.allegaert@uzleuven.be (K.A.); 3Pharmacy Department, University Hospitals Leuven, B-3000 Leuven, Belgium; 4Department of Laboratory Medicine, University Hospitals Leuven, B-3000 Leuven, Belgium; stefanie.desmet@uzleuven.be; 5Department of Microbiology, Immunology and Transplantation, KU Leuven, B-3000 Leuven, Belgium; 6Department of Development and Regeneration, KU Leuven, B-3000 Leuven, Belgium; willem-jan.metsemakers@uzleuven.be; 7Department of Trauma Surgery, University Hospitals Leuven, B-3000 Leuven, Belgium; 8Department of Clinical Pharmacy, Erasmus MC, NL-3015 CN Rotterdam, The Netherlands

**Keywords:** antibiotic, clindamycin, bacterial infections, pharmacokinetics, special patient populations, CYP450 enzymes, drug–drug interactions

## Abstract

Given the increase in bacterial resistance and the decrease in the development of new antibiotics, the appropriate use of old antimicrobials has become even more compulsory. Clindamycin is a lincosamide antibiotic approved for adults and children as a drug of choice for systemic treatment of staphylococcal, streptococcal, and gram-positive anaerobic bacterial infections. Because of its profile and high bioavailability, it is commonly used as part of an oral multimodal alternative for prolonged parenteral antibiotic regimens, e.g., to treat bone and joint or prosthesis-related infections. Clindamycin is also frequently used for (surgical) prophylaxis in the event of beta-lactam allergy. Special populations (pediatrics, pregnant women) have altered cytochrome P450 (CYP)3A4 activity. As clindamycin is metabolized by the CYP3A4/5 enzymes to bioactive N-demethyl and sulfoxide metabolites, knowledge of the potential relevance of the drug’s metabolites and disposition in special populations is of interest. Furthermore, drug–drug interactions derived from CYP3A4 inducers and inhibitors, and the data on the impact of the disease state on the CYP system, are still limited. This narrative review provides a detailed survey of the currently available literature on pharmacology and pharmacokinetics and identifies knowledge gaps (special patient population, drug–drug, and drug–disease interactions) to describe a research strategy for precision medicine.

## 1. Introduction

The widespread use of antibiotics has contributed to the ongoing increase in antimicrobial drug resistance over the last years, an emerging health issue highlighted as a key research area of the Health Research and Innovation by the European Commission [1]. This worldwide health problem [2], followed by the decrease in the development of new antimicrobials, resulted in a worrying situation, as alternatives to treat infectious diseases become scarce. Given this concern, new approaches are needed, including the appropriate use of old ‘wonder drugs’ such as clindamycin (CLI), a potent and widely used antimicrobial. CLI is currently being exploited for its use in combination with other agents for combating resistant pathogens. In addition, CLI’s systemic exposure depends on drug hepatic clearance and drug–drug interactions that may occur and are of interest for reference and special patient populations. In addition, drug–disease interactions may also occur in the presence of inflammation or disease state. Subsequently, in this narrative review, we will provide a detailed survey of the currently available literature on the pharmacology and pharmacokinetics of CLI, and we will identify relevant knowledge gaps (special patient population, drug–drug, and drug–disease interactions) to describe a future research approach towards personalized medicine (Figure 1).

CLI (7-chloro-7-deoxy-lincomycin, CLI) is a lincosamide antibiotic derived from lincomycin, naturally produced by *Streptomyces lincolnensis*. This semisynthetic chlorinated derivative (Figure 2, Table 1) was developed in the mid-1960′s and is currently the main lincosamide antibiotic applied in daily clinical practice. Lincosamides are a relatively limited antimicrobial class with a unique composition of sugar and amino acid moieties [3]. CLI is a drug approved by the U.S. Food and Drug Administration (FDA) for use in adults and children requiring systemic treatment of staphylococcal, streptococcal, and anaerobic bacterial infections. It is most often used for treating beta-lactam-allergic patients or in other situations where beta-lactams cannot be used.

CLI is available in three forms: CLI hydrochloride salt (oral capsules), CLI palmitate hydrochloride salt (oral suspension), and CLI phosphate (injection solution) esters. Systemic infections can be treated orally (PO) or intravenously (IV) (Figure 2, Table 1 and Table 2). There are currently three marketed formulations available for these routes of administration (Table 2): capsules (75 mg, 150 mg, 300 mg) and a solution (75 mg/5 mL) for PO administration, and a 150 mg/mL IV solution for injection or infusion. IV formulations are available as CLI phosphate (300 mg/2 mL, 600 mg/4 mL, 900 mg/6 mL), and CLI phosphate in D5W or NaCl 0.9% (300 mg/50 mL, 600 mg/50 mL, 900 mg/50 mL).

## 2. Pharmacology and Target

Standard dosing of CLI used in a healthy adult population—assuming a median adult 70 kg standardized total body weight (TBW)—is 600 mg.q8h PO or IV, as is recommended in clinical practice guidelines for community-acquired methicillin-resistant strains of *Staphylococcus aureus* (MRSA) infections [4]. For more severe infections, the Summary of Product Characteristics (SmPC) recommends an IV dosing of 1200 to 2700 mg/day in 2–4 equal doses, while IV doses up to 4800 mg/day have been given in adults with life-threatening infections.

Concentrations of CLI in the serum increase linearly with increased dose, and exposure exceeds the minimum inhibitory concentration (MIC) for most indicated organisms for at least 6 h following administration of the recommended dose [5]. Hereby, the ratio between the 24 h area-under-the-serum-concentration-time curve (AUC) for the free or unbound drug (f) concentration and the current MIC ([fAUC0-24h/MIC]) is the most important PK/PD parameter [6,7,8] to predict CLI’s clinical efficacy. CLI exerts outstanding activity—both in vitro and in clinical infections—against aerobic Gram-positive cocci (e.g., *Staphylococcus* spp., *Streptococcus* spp.) and a wide range of anaerobic bacteria (e.g., *Fusobacterium* spp., *Prevotella* spp., *Bacteroides* spp.). Aerobic Gram-negative bacteria are not usually susceptible to this antibiotic. Relevant cases of *Staphylococcus* spp. Relate to methicillin-susceptible strains of *Staphylococcus aureus* (MSSA). In addition, MRSA and coagulase-negative staphylococci (CoNS) (e.g., *Staphylococcus epidermis, Staphylococcus haemolyticus*) are frequently resistant to CLI [9]. MIC values (clinical breakpoints) for CLI’s most relevant bacterial pathogen strains—regarding further discussed systemic infections—are interpreted using the European Committee on Antimicrobial Susceptibility Testing (EUCAST) 2022 interpretation guideline [10] (Table 3).

In susceptible organisms, CLI directly inhibits peptide bond formation as it specifically binds to the 23S ribosomal RNA of the 50S subunit of the bacterial ribosome by affecting the process of peptide chain initiation. Hence, dissociation of peptidyl-tRNAs from ribosomes is most likely stimulated [11]. Bacteriostatic activity occurs in most situations [12], as the binding inhibits the active site of the ribosome unit and interrupts the early stages of protein synthesis. However, at higher drug concentrations, CLI can exhibit a time-dependent bactericidal action against sensitive strains, killing the bacteria with a postantibiotic effect [13]. Given this common mode of action (Table 2), CLI resistance is primarily due to the expression of a methylase genes (*Erm*) that lead to methylation of the 23S ribosomal RNA-binding site (so-called MLS_B_ resistance) [14,15]. Resistance to CLI can also be caused by modification/inactivation of the antibiotics by specific enzymes or active efflux from the bacterial cell [11]. Cross-resistance mechanism to macrolides, lincosamides, and streptogramin B (MLS_B_) is frequent in *Staphylococcus* spp. (MSSA and MRSA) and *Streptococcus* spp. Erythromycin is an effective macrolide inducer of such resistance. Given the variability of MLS_B_ resistance to CLI, the disk induction test (D-test) is recommended in practice for its detection in erythromycin-resistant and CLI-susceptible isolates [9,16].

## 3. Pharmacokinetics

### 3.1. Pharmacokinetics in Reference Population

Pharmacokinetic (PK) properties of antibiotics are mainly based on their chemical structure, which absolutely affects their bioavailability, half-life, tissue penetration, distribution, degradation, and elimination [17]. CLI can be administered by IV route for systemic infections, but the PO route is also possible, as the drug shows rapid and extensive absorption from the gastrointestinal (GI) tract with a bioavailability of ~87% (estimated value of 87.6% ± 0.09, from a population PK study [18]). Note that observed bioavailability in healthy subjects and patients with acquired immunodeficiency syndrome (AIDS) ranged from 53 to 75% [19]. In adults given a standard dose of 150 mg PO, an average peak serum CLI level of 2.5 mcg/mL is achieved within 45 min [20]. It appears that CLI can be transported both by passive diffusion and by nucleoside transporters [21]. This conclusion is based on CLI’s ability to compete with adenosine for uptake into phagocytes [22]. Additionally, the PO administered dose is absorbed into the intestinal epithelial cells without appreciated variations of drug concentration during concomitant food administration. From a theoretical point of view, in case the GI tract of the patient is intact, and the bioavailability of an oral antibiotic agent (CLI) is adequate, it should be possible to reach sufficient antibiotic exposure with PO administered antibiotics (CLI) [23]. However, the systemic response to infection might influence the PKs of antibiotics. To date, data on the bioavailability of PO-administered antibiotics during the initial phase of a systemic infection in non-ICU hospitalized patients are scarce and contradictory [23]. That is to say, we do not know whether adequate antibiotic (CLI) levels can be reached in the systemic circulation when such drugs are administered PO during the initial stage of an infectious illness [23].

Given its lipophilic properties, CLI has a high volume of distribution (Vd) (in healthy adults, CLI has a Vd at a steady state of 0.79 L/kg [19]). The degree of protein binding in healthy adults is concentration-dependent and ranges from 62% to 94%. Binding relates primarily to alpha-1-acid glycoprotein (AAG) concentration [8], followed by plasma protein albumin. As it travels through the bloodstream, this lincosamide is widely distributed in body fluids, organs, and tissues, including bones and abscesses. CLI crosses the human placenta readily, but it does not efficiently cross the blood–brain barrier, and no significant levels (~20%) [13] are attained in the cerebrospinal fluid, even in the setting of meningitis [20,24,25]. As described in the SmPC, high CLI serum concentrations are achieved in bone tissue, synovial fluid, peritoneal fluid, pleural fluid, expectorations, and pus. The following concurrent concentrations of the drug are reported compared with the blood compartment: in bone tissue 40% (20–75%), in synovial fluid 50%, in peritoneal fluid 50%, in pleural fluid 50–90%, in expectorations 30–75%, and in pus 30%. In a reference population, CLI exhibits good bioavailability after PO administration [18].

CLI undergoes hepatic metabolism to the major bioactive sulfoxide (primary metabolite) and N-demethyl metabolites (Figure 2, Table 1), and also some inactive metabolites. Recent in vitro studies in human liver and intestinal microsomes indicated that CLI is predominantly oxidized by phase I cytochrome P450 (CYP)3A4 enzyme, with a minor contribution from phase I CYP3A5, to form the aforementioned metabolites [5]. In healthy adults, hepatic drug clearance (CL) is about 0.3–0.4 L/h/kg [5]. Because CLI has a low hepatic extraction ratio, CL decreases when hepatic intrinsic clearance and the unbound concentration decrease [19].

After hepatic metabolism and within 24 h after intake, only about 10% of an oral dose of CLI is excreted in the urine as active drug and metabolites, ~3.6% in the feces, with the remainder excreted as inactive metabolites [26]. Biological elimination half-life (T_1/2_) in healthy adults with a normal renal function is about 2–3 h [26]. The T_1/2_ may be prolonged in patients with moderate to severe hepatic or renal dysfunction, but no specific dosing adjustment is recommended. Dosage adjustment and monitoring are only recommended for patients with severe hepatic impairment or failure, but there are no specific guidelines for CLI. Since CLI is hepatically cleared, adjustment for renal dysfunction is generally not required.

### 3.2. Pharmacokinetics in Special Patient Populations

CLI’s metabolites are not just of scientific interest but can also be relevant when considering bacterial killing. Better understanding and quantitative data of their potential pharmacodynamic (PD) activity are still missing. Given the biotransformation pathway involved, it is important to elucidate relevant covariates of drug metabolism, such as age, gestation, or drug–drug interaction, as these hold the potential for individual exposure and response to therapy. Subsequently, extrapolating the conventional adult dose of CLI based on covariates such as TBW is not appropriate and should be further explored.

Special patient populations (SPPs) undergo several alterations that can potentially impact both the metabolism and disposition of CLI. We focused on two SPPs (pediatrics and pregnancy, breastfeeding, and postpartum) to illustrate the needed insights and research to improve precision medicine in these populations (Table 4). Anatomic and physiological age-related changes tend to be related to TBW, body composition, and function (e.g., variations in fat mass, body water, plasma volume and proteins, and glomerular filtration). These changes can be further affected by pathophysiological and nonmaturational events such as inflammation. One of the relevant mechanisms of altered PK in these special populations is due to variations in the CYP3A activity. Consequently, knowledge of the potential relevance of CLI’s metabolites and disposition in SPPs is fundamental. We selected these two SPPs, but the same strategy could be considered for other specific settings, such as for patients with cancer or coronavirus disease 2019 (COVID-19).

Pediatrics is a diverse SPP as it covers pediatric age groups, including neonates (birth to <28 days), infants (28 days to <2 years), young children (2 to <6 years), old children (6 to <12 years), and adolescents (12 to 18 years), as defined by the International Conference on Harmonization E11 guidance (International Council for Harmonisation, 2000) [27]. The second SPP group includes pregnant, breastfeeding, and postpartum women. Pregnancy changes are not uniform, and their extent depends on the stage of gestation; hereby, this SPP is divided into five classes of women: prepregnant, first trimester, second trimester, third trimester, and breastfeeding women or postpartum women. In order to guarantee safe and efficient use of CLI in each SPP, its PK data should be considered, as well as dosage guidelines concerning the systemic treatment of relevant bacterial infections. Many of the anatomic and physiologic changes observed in these populations will result in marked changes in absorption, distribution, metabolism, elimination (ADME) [8] (Table 4), and dosing regimens. 

#### 3.2.1. Pediatrics

CLI has been used for many years in the treatment of infections in children. It has traditionally been a component of empiric antibiotic regimens for bone and joint infections where anaerobes are likely causative pathogens, as well as in the treatment of serious skin and soft tissue infections. The exact ontogeny of oral drug absorption GI processes in pediatrics is still to be elucidated, especially for the most vulnerable groups (neonates and infants of <6 months). Consequently, drug labeling changes frequently and does not include neonatal dosing data (<1 month) [28]. In neonates, the dose should be based on both TBW and age to allow for a slower elimination [20]. According to the SmPC, for the treatment of serious infections, CLI 15 to 25 mg/kg/day can be administered (in three or four equal doses) for children and adolescents. For more severe infections, doses of 25 to 40 mg/kg/day for children and adolescents can also be administered in three or four divided doses. Despite the existing guidelines, available PK data from pediatric PK studies are scarce, and age-based optimal dosing is still unknown [29]. It is recommended that children be given no less than 300 mg/day regardless of the TBW.

Apart from a brief peak postnatally (pH ~7), where mean gastric pH is high directly after birth, gastric pH rapidly decreases (pH ~2.0–2.7 values in neonates [30]) and remains around a value of 2–3 in children of all ages [31] (typically < 3 in children and adolescents [30]). As CLI shows high lipophilicity, changes in gastric pH do not translate into absorption rate variations [32]. However, data concerning the extent to which absorption in the GI tract could be affected for neonates are deficient: gastric emptying reported in the literature is highly variable in children younger than 6 months [33], and GI transit time is also slower in neonates and infants yet reaches adult values at the age of 2 years [33,34] (Table 4). Nevertheless, CLI is a highly bioavailable PO, but, given the clear knowledge gap regarding the bioavailability of PO administered antibiotics in non-ICU patients during the initial phase of systemic infection [23], CLI can also be administered by IV route for systemic infections.

Distribution in pediatrics [29] can be notably affected by changes in protein binding (Table 4). Neonates display a continually changing plasma profile, as the presence of fetal proteins and endogenous substrates, which are known to interfere with drug binding, can lead to unexpected complications because of a higher than expected ‘free’ drug fraction. Moreover, serum protein level and drug-binding rate of newborn infants are very low, and adult values (77 mg/mL) are reached at ~10 months of age [35]. Besides maturational changes, AAG concentrations are also affected by nonmaturational factors such as postsurgery or inflammation, as both result in an increase in AAG synthesis as part of the acute-phase response. Booker et al. determined how concentrations of AAG changed in infants requiring major surgery [35]. Despite a high interpatient variation (0.07–0.78 mg/mL), the overall mean perioperative AAG was 0.38 mg/mL, although concentrations doubled to 0.76 mg/mL on day 4 after surgery. Next to AAG’s response to surgical stress, inflammation in disease states can impact AAG levels in pediatrics. İpek et al. confirmed results from previous studies by reporting a statistically significant increase in AAG levels during neonatal bacterial sepsis, showing positive values (>0.7 g/L) of 1.1 ± 0.4 and 0.8 ± 0.4 g/L for confirmed and clinical sepsis, respectively [36].

Regarding developmental changes in CLI’s hepatic metabolism during childhood, CYP3A4 is not detectable before birth. However, CYP3A4 generally increases postnatally to become the dominant CYP enzyme in the adult liver and intestine [37] and becomes active during the first weeks [38]. A recently published review from van Groen et al. [39] stated an increase in CYP3A4 microsomal levels, although activity data show a decrease during fetal life (from 6% to 3%). CYP3A4 enzyme shows low catalytic activity in fetuses with 50% of adult levels in infants up to 1 year, while adult levels are slowly reached in infants and young children (between 1 to 5 years). Subsequently, CYP3A4′s enzyme activity is found to be age-dependent (Table 4). In terms of CYP3A5, van Groen et al. stated that protein expression shows no clear developmental pattern and is age-independent [32,39]. In contrast, CYP3A5 is polymorphic, and its expression levels vary between individuals and populations. CYP3A5 can be found in significant levels in 10–40% of Caucasians, 40–50% of Chinese subjects, and in approximately 90% of individuals with African origins [40].

Besides maturation, inflammation and organ failure are proven to be relevant nonmaturational factors for CYP3A4-mediated drug metabolism. Brussee et al. [41] recently carried out the first population PK model that quantified the influence of maturation, inflammation, and organ failure on midazolam CL (and potentially other selective CYP3A substrates) in term neonates, infants, children, adolescents, and adults with varying levels of critical illness. Predictions based on this model indicated a 30% decrease in midazolam CL when C-reactive protein concentrations that reflect the presence of inflammation threefold increase from 32 to 100 mg/L [41]. Furthermore, CL decreased by 26% when disease severity, expressed as the number of failing organs, increased from one to two [41].

**Table 4 antibiotics-11-00701-t004:** Pharmacokinetics in the specified special patient populations.

	Pediatrics					Pregnant, Breastfeeding and Postpartum Women
PK Covariates	Neonates (0 to <28 Days)	Infants (28 Days to <2 Years)	Young Children (2 to <6 Years)	Old Children (6 to <12 Years)	Adolescents (12 to 18 Years)	
**Absorption**						
Gastric pH ^2^	pH ~7: postnatal peak pH ~2.0–2.7: rapid decrease after birth	pH ~2–3	pH ~2–3	pH ~2–3	pH ~2–3	Pregnant women: increased gastric acid secretion, but no major changes in gastric pH
Gastric emptying	Highly variable	Highly variable until ~6 months	More stable	More stable	More stable	Pregnant women: gastric emptying does not appear to be affected
GI ^1^ transit time	Slower than adults	Slower than adults	Adult values	Adult values	Adult values	Pregnant women: GI transit time could be longer in the third trimester when intestinal motility is lower
Other factors	NS ^1^					Pregnant women: nausea and vomiting also diminish absorption in the early pregnancy
**Distribution**						
Protein binding: maturational changes ^3^	Low protein binding rate ^7^	Low protein binding rate until ~10 months, then adult value rate	Adult value rate (77 mg/mL)	Adult value rate (77 mg/mL)	Adult value rate (77 mg/mL)	Pregnant women: reduction in AAG ^1^ and albumin fractions over pregnancy trimesters. From ~100% (prepregnant, first trimester) to ~80% (second, third trimesters) [8]
Protein binding: nonmaturational changes ^4^	Generally increase in serum AAG concentrations					Type of delivery (cesarean or vaginal): increase in AAG serum concentrations, no significant changes in albumin
Transplacental distribution	NA ^1^					Breastfeeding women: human breast milk concentrations of ~0.7 – 3.8 mcg/mL during lactation
**Metabolism**						
CYP3A4 ^1^ enzyme activity ^5^	Postnatal increase in microsomal levels ^8^, 50% of adult levels	50% of adult levels until 1 year, then adult values are slowly reached	Adult values are slowly reached (between 1 to 5 years)	Adult values	Adult values	Pregnant women: drastic increase in CYP3A4 enzyme activity from prepregnant (~100%) to first, second and third trimester (~210%) [8]
Drug CL ^1^ CYP3A4-substrate (midazolam) ^6^	Results from a popPK ^1^ model quantifying CL changes [34]: 30% decrease in midazolam CL in the presence of increasing inflammation (3-fold), 26% decrease in midazolam CL in the presence of increasing organ failure (from 1 organ to 2)					Pregnant women: ~100-fold increase in sex hormones. Increase of CL [35] for pregnant women (593 ± 237 L/min) compared with postpartum (343 ± 103 L/min)

^1^ GI—gastrointestinal; NS:—not specified; NA—not applicable; CYP3A4—cytochrome P450 (CYP)3A4; AAG—alpha-1-acid glycoprotein; CL—clearance; popPK—population pharmacokinetics; ^2^ As clindamycin shows high lipopylicity, changes in gastric pH do not translate into absorption rate variations; ^3^ Maturational changes refer to developmental variations during aging; ^4^ Nonmaturational changes refer to, e.g., postsurgery (surgical stress) and inflammation during chronic disease states (organ failure) or as part of the acute phase response; ^5^ Both CYP3A4 enzyme activity and expression are found to be age-dependent, while expression of CYP3A5 isoenzyme shows no clear developmental pattern and is age-independent; ^6^ Drug CL changes are observed in the CYP3A4-substrate midazolam, used as probe drug for measurement; ^7^ Neonates display a continually changing plasma profile (i.e., fetal proteins, endogenous substrates) that can interfere to protein binding, causing a higher than expected ‘free’ drug fraction and leading to unexpected complications; ^8^ Despite of the postnatal increase in CYP3A4 microsomal levels, activity data show a decrease during fetal life (from 6% to 3%).

#### 3.2.2. Pregnant, Breastfeeding, and Postpartum Women

CLI for systemic use is FDA-classified as a Pregnancy Category B drug and is generally considered safe and effective in pregnancy [26], given that, based on experimental animal studies, CLI is not expected to increase the risk of congenital anomalies [42]. Because risk cannot be excluded, the SmPC states that the drug should be used during pregnancy only if clearly needed. According to the lactation section on the SmPC, PO-, and IV-administered CLI diffuses across the placenta barrier into the fetal circulation, appearing in human breast milk in ranges from 0.7 to 3.8 mcg/mL (i.e., breast milk/maternal plasma ratio 0.08–3.1 [8,43]). Hence, nursing mothers should stop breastfeeding during CLI therapy (if possible) [42] because CLI has the potential to cause undesired effects on breastfed infants’ GI flora. Moreover, CLI should not be taken by nursing mothers because of the potential for serious adverse reactions in nursing infants [42].

The effect of pregnancy on drug absorption over trimesters is mainly determined by factors such as gastric pH, gastric emptying, and GI transit time (Table 4). Previous reviews [44,45] reported increased gastric acid secretion during pregnancy, but several original studies addressing heartburn during pregnancy show that there are no major changes in gastric pH over trimesters compared with nonpregnant women [33]. Gastric emptying does not appear to be affected by pregnancy [33,46], but overall, GI transit time could be longer in the third trimester when intestinal motility is lower [33]. Drug absorption is also diminished by nausea and vomiting early in pregnancy, which can result in lower plasma drug concentrations [44].

As mentioned earlier, significant levels of CLI are achieved in human breast milk as a result of its distribution across the blood–placental barrier (Table 4). This distribution during pregnancy can be influenced by changes in TBW, regional blood flows, tissue composition (such as body water and body fat), plasma composition, and volume and alterations in the unbound fraction of a given antimicrobial [8]. Over the pregnancy trimesters, most covariates (e.g., TBW, fat mass, body water, plasma volume, cardiac output) increase until the third trimester. However, both AAG and albumin fractions are reduced from ~100% (prepregnant and first trimester) to ~80% (second and third trimesters), as described by Allegaert et al. [8] (Table 4). Consequently, the binding of CLI may be affected, which may lead to difficulties in maintaining adequate plasma concentrations of CLI (highly protein-bound) as the measurement of total drug concentration in plasma may no longer be a valid indicator for dose adjustment. As studied by Larijani et al. [47], the type of delivery (cesarean or vaginal) further affects serum AAG and albumin concentrations. Regarding AAG levels, study findings demonstrate that in both the cesarian section and tuboplasty patients, exposure increased similarly on different postoperative days (146.3 ± 19.2; 134.8 ± 8.2 mg/dL). Moreover, even if the increase in AAG was delayed in pregnant women, both vaginal and cesarean delivery resulted in an increase in serum AAG concentration. Concerning albumin levels, data indicated that nonpregnant women had higher serum albumin concentrations than pregnant women. However, albumin concentrations did not significantly change throughout the study except for a significant immediate decrease postoperative after the cesarian section.

Pregnancy is characterized by about a 100-fold increase in circulating progesterone and estrogens, with an increase over advancing gestational age [48]. Subsequently, drug metabolism is also altered over pregnancy, secondary to the characteristic elevated levels of sex hormones and changes in drug-metabolizing enzymes [44]. Hebert et al. carried out a study to evaluate the effects of pregnancy on in vivo CYP3A activities in humans using CYP3A4-substrate midazolam as a probe drug for measurement [49] (Table 4). They reported a higher (apparent) oral unbound drug CL for pregnant women (593 ± 237 L/min) compared with postpartum (343 ± 103 L/min), representing approximately twofold CYP3A4 in vivo induction during the third trimester [49]. Later findings confirmed that placental growth hormone estrogens (17-beta estradiol), cortisol, and progesterone potentially induce this in vivo CYP3A4 increase [50]. Moreover, Allegaert et al. reported a drastic increase in CYP3A4 enzyme activity from the prepregnant stage (100%) to the first, second and third trimesters (210%), next to increasing RNA expression protein levels and hepatic CL [8].

## 4. Clinical Practice and Efficacy

CLI is FDA-approved for serious bacterial infections, mainly caused by *S. aureus*, such as septicemia, intra-abdominal infections, lower respiratory tract infections, bone and joint infections, and skin and soft tissue infections [51]. Anesthesiologists and surgeons will often administer CLI per The American Society of Health-System Pharmacists (ASHP) and Infectious Diseases Society of America (IDSA) guidelines as prophylaxis in the operating room [51]. Evidence for the use of this drug in systemic conditions is reviewed to show CLI’s potential applicability and indications (Table 2 and Table 5). Moreover, recommended MIC values (clinical breakpoints) for CLI’s most relevant pathogen strains for the mentioned indications are interpreted using the EUCAST 2022 guideline (Table 3).

### 4.1. Surgical Prophylaxis in the Event of Beta-Lactam Allergy

According to the European Centre for Disease Prevention and Control (ECDC), perioperative antibiotic prophylaxis is considered one of the most effective measures to prevent surgical site infections (SSIs) [53]. SSIs are considered one of the most common postoperative complications and significantly contribute to patient morbidity and mortality in hospital settings [54,55], taking place within 30 days after the operation [53]. The ASHP guideline on Antimicrobial Prophylaxis in Surgery describes that the predominant organisms causing SSIs after clean procedures are mostly *S. aureus*, followed by CoNS. In clean-contaminated procedures, including abdominal procedures and heart, kidney, and liver transplantations, the predominant pathogens include Gram-negative rods and *Enterococcus* spp. in addition to skin flora. Infections caused by *Enterococcus* spp. cannot be treated with CLI, as the lincosamide does not cover them. Although not a first-line treatment, the Surgical Infection Society (SIS) has published guidelines for using CLI as an alternative for surgical antimicrobial prophylaxis in patients with allergies to beta-lactam drugs such as cefazolin, considered the agent of choice for SSIs [56]. The SIS guidelines, focused on primary perioperative prophylaxis, were developed jointly by the ASHP, the IDSA, the SIS, and the Society for Healthcare Epidemiology of America (SHEA).

Depending on the SSI and its severity, CLI can be administered as monotherapy or combined with other antimicrobials (e.g., aminoglycosides, fluoroquinolones, etc.), and the preferred route of administration varies with the type of procedure. For the majority of procedures, IV administration is ideal as it produces rapid, reliable, and predictable serum and tissue concentrations [56]. Regarding the SIS guidelines [56], standard preoperative IV dosing in adults and pediatrics is 900 mg and 10 mg/kg, respectively, and continued every 6 h. This means that from the initiation of the preoperative dose, redosing every 6 h is recommended, as long as the indication of prophylaxis holds. However, practical guidelines provided by the MDA Anderson Cancer Center and Sarasota Memorial Health Care System [57,58] recommend an IV weight-based dosing of 600 mg (<70 kg) and 900 mg (≥70 kg) every 6–8 h, depending on the center. Standard IV dosing recommendation of 600 mg.q8h (<70 kg) is also followed by the University Hospitals Leuven, among other European facilities.

### 4.2. Prophylaxis and Treatment of Pregnancy Infections

CLI has been extensively prescribed for several decades to prevent or treat infections during pregnancy and peripartum [8,26]. The most commonly performed surgical procedure during pregnancy is cesarean delivery, which may lead to postcesarean infections (e.g., maternal febrile morbidity, wound infection, endometritis, and other serious complications) [8,59]. Other pregnancy infections, such as endometritis or chorioamnionitis, can occur after fetal surgery [8]. In the presence of maternal allergy to penicillins, CLI is a potent alternative for the antimicrobial prophylaxis and treatment of pregnancy and peripartum infections caused by *S. aureus*, followed by CoNS and group B *Streptococcus* (GBS) in the event of, e.g., allergy to beta-lactams [60].

Recommended guidelines produced by the Center for Disease Control (CDC) and the American College of Obstetricians and Gynecologists recommended a CLI dosage of 900 mg.q8h IV until delivery in the case of prophylaxis for GBS neonatal disease [61,62], resulting in a rapid decline of colony counts (<5%) within the first 2 h of administration [8]. In women during pregnancy or postpartum period, genital tract colonization with GBS is usually asymptomatic, but GBS clinical manifestations include the abovementioned infections [63]. Furthermore, data from a prospective study on the transplacental passage of CLI confirmed that the CDC-recommended dosage produces therapeutic maternal and cord blood levels [43].

### 4.3. Treatment of Diabetic Foot Infections

A diabetic foot infection (DFI) is defined as any inframalleolar infection in a person with diabetes mellitus and mostly arises from diabetic foot ulcers [64]. DFIs are considered specific skin and soft tissue infections that, provided by the severity of the lesion or a foot ulcer, can cause complications (e.g., diabetic foot osteomyelitis), leading to morbidity, hospitalization, and amputations. According to infection severity (wounds), the 2012 IDSA Guideline for the Diagnosis and Treatment of DFIs [65] classified the infections into mild (superficial and limited size and depth), moderate (deeper or more extensive), or severe (accompanied by systemic signs or metabolic perturbations). Commonly isolated microorganisms are *S. aureus*, beta-hemolytic streptococci, and Gram-negative species [66]. Acute infections are usually monomicrobial, while serious infections in hospitalized patients are often caused by 3–5 bacterial species [64,67].

According to the 2012 IDSA guideline, treatment with CLI is an effective choice for treating mild, moderate, and severe DFIs. Although not directly stated in the IDSA 2012 guidelines, CLI is not usually administered as a first-line treatment but as an alternative in the event of severe beta-lactam allergy [68,69]. In the case of mild DFIs caused by MSSA and *Streptococcus* spp., PO administration of CLI (300–450 mg.q8h) is recommended [65,70]. When systemic signs emerge, moderate to severe coinfections may require hospitalization and can be temporarily treated by IV coadministration of CLI (600 mg.q8h is the dosing regimen used by University Hospitals Leuven) with ciprofloxacin or levofloxacin until patient stabilization, switching then to the oral equivalent [65,69]. CLI is a good choice given its high oral bioavailability and bone penetration profile. This lincosamide has also shown to be effective in clinical trials, including in patients with DFIs and in the treatment of diabetic foot osteomyelitis [65,69,71], an infectious disease that is one of the most common expressions of DFIs. However, CLI is not currently an antimicrobial specifically approved by the FDA for the treatment of DFIs [65].

### 4.4. Treatment of Bone and Joint, Fracture-Related, and Periprosthetic Joint Infections

Bone and joint infections (BJIs) can be subdivided into multiple subgroups, including fracture-related infections (FRIs), periprosthetic joint infections (PJIs), spinal infections, septic arthritis, and diabetic foot osteomyelitis [52,72]. PJI is a serious complication after total joint arthroplasty that remains a core problem in orthopedic surgery [73] and is considered a device-associated infection. FRI, on the other hand, also remains a challenging complication that primarily creates a heavy burden for orthopedic trauma patients. Compared with PJIs, FRIs have unique features (i.e., fracture, bone healing, soft tissue injury) that need to be considered [74]. The most common (biofilm-forming) bacteria causing BJIs in adults are *Staphylococcus* spp. (MSSA, MRSA), followed by CoNS, *Streptococcus* spp., *Enterococcus* spp., *Pseudomonas aeruginosa*, and anaerobic bacteria [52,69]. Few studies have reported cases with well-identified *Streptococcus* spp. After conducting a 5-year study of interregional reference centers in the south of France, Seng et al. [75] reported the five most represented *Streptococcus* spp. to be GBS or *S. agalactiae* (37%), *S. dysgalactieae* (12%), *S. anginosus* (11%), *S. constellatus* (10%), and *S. pneumoniae* (9%). In contrast, CLI is not active against *Enterococcus* spp.

BJIs treatment generally requires surgery combined with antibiotic therapy. Even if the duration of antimicrobial therapy in BJIs is controversial and not well investigated, antibiotic therapy is normally IV administered for at least 6 weeks and continued with PO administration for up to 12 weeks in case an implant is present [52,74]. Moreover, a recent randomized controlled trial showed that patients treated with up to 7 days of IV antibiotics followed by oral therapy had the same outcome as those with prolonged IV therapy (usually 6–12 weeks) [74,76]. The usual CLI dosage used in adults for the treatment of BJIs is 600 mg.q8h taken PO or IV [18]. CLI is widely used for the treatment of BJIs in adults and children for its potential activity against biofilm-forming bacteria and high levels of bone penetration of 30% [18,77,78]. In addition, CLI has good oral bioavailability and is well tolerated [19,79]. A Consensus from an International Expert Group [52] published recommendations for systemic antimicrobial therapy in FRI, with variations in CLI’s dosing regimen depending on the causative infectious agent.

To prevent the emergence of *S. aureus* resistance, synergistic bactericidal activity at higher concentrations has been achieved through the concomitant administration of CLI and rifampicin (RIF-CLI). Rifampicin-based combination therapy regimens have been shown effective in eradicating staphylococcal biofilms [80,81]. Several guidelines have been published regarding the use of antibiotic therapy in conjunction with surgery when necessary [82,83], but no specified regulations have been established for this RIF-CLI combination, as due to the involvement of the CYP3A4 metabolism, a common fear of drug–drug interactions have been an ongoing research question. Thus, most aspects of antibiotic treatment for BJIs are still mostly based on expert opinions. Recommendations by the Consensus from an International Expert Group suggest coadministration of CLI (600 mg.q8h) with rifampicin (300–450 mg.q12h) to treat systemic FRIs caused by *Staphylococcus* spp. [52]. Overall, this specific combination (RIF-CLI) shows potential clinical applicability for BIJs and is discussed later in the review.

Bone composition is different from that of the other tissues, such as cardiac muscle or lung tissue, because bone and joint tissues are less vascularized. It is, therefore, difficult to predict whether agents showing good penetration into other tissues will also achieve high concentrations in bone [84]. Consequently, knowledge of the rate and extent of antibiotic penetration into bone tissue is fundamental for the successful treatment of BJIs.

CLI’s high levels of bone and joint penetration (~30%) are due to its physicochemical and PK properties. A bone concentration/MIC ratio of 5 is required for time-dependent bactericidal action [85]. CLI is suitable for this clinical purpose because of its good penetration profile and adequate activity (AUC/MIC ratio) against biofilm-forming bacteria. Most bone penetration studies of CLI were conducted in the 1970s [84,86]. CLI concentrations were determined by bioassay, and the findings of these studies display substantial variability in the reported mean bone/plasma concentration ratio of CLI (0.21–0.45). This could be due to the presence of an infection, differences in the analytical techniques applied, or active metabolites of CLI measured in the bioassay [84]. Quantification of drugs by bioassay was carried out in these older studies, whereas nowadays, LC-MS methods are considered much more sensitive and accurate. A review published by Thabit et al. [87] on antibiotic bone penetration indicates that CLI could reach the susceptibility MIC breakpoint of Gram-positive cocci in ischemic tissues (≤0.25 mg/L for *Streptococcus* spp. and ≤0.5 mg/L for *Staphylococcus* spp.), but less likely that of anaerobes [(4)^3^ mg/L for *Bacteroides* spp.]. MIC clinical breakpoint values have been updated using the EUCAST 2022 interpretation guideline [88].

Rifampicin also has an optimal penetration into biofilms compared with other currently available antimicrobial agents but cannot be administered in monotherapy because of the rapid emergence of resistant mutants [77]. Subsequently, RIF-CLI is a well-tolerated combination [77] that has been tested for the treatment of BJIs. Both antimicrobials are inexpensive, possess a good oral bioavailability, and can successfully infiltrate the rigid bone structure into the synovial space to inhibit biofilm formation and bacterial adherence [18,77,78]. In 2012, Bouazza et al. [18] performed a retrospective population PK study to predict optimal administered PO and IV dosing of CLI for patients with osteomyelitis. Results indicated that CLI’s CL increases with TBW, so presumably, standard doses (600 mg.q8h) should be incremented (900 mg.q8h) for patients over the ~70 kg standardized TBW. Moreover, this study indicated a potential effect of RIF-CLI on the CL, as CLI’s CL was increased by 43%, and subtherapeutic CLI concentrations were observed under combined therapy [18]. These assumptions should be prospectively confirmed. Thus, a few prospective PK studies have been conducted over the past decade to improve the treatment of BJIs (Table 6) [79,85,89]. All these studies reported plasma concentrations in order to estimate CLI’s penetration into the bone during monotherapy and in combination (RIF-CLI), generally administered PO and IV.

In 2015, Curis et al. [79] prospectively analyzed the influence of rifampicin on CLI’s plasma concentrations in patients with BJIs, confirming the previous hypothesis that indicated a slight increase in treatment failure associated with increased TBW. Findings also confirmed that CLI-RIF administration lowers the trough concentrations (C_min_) and peak concentrations (C_max_) of CLI. CLI measured plasma concentrations (monotherapy vs. combined treatment) resulted in median C_min_ (1.36 vs. 0.29 mg/L) and median C_max_ (7.48 vs. 4.46 mg/L) values that were also lower for patients under CLI-RIF treatment. In addition, Bernard et al. [89] (2015) reported mean C_min_ (4.7 vs. 0.79 mg/L) and mean C_max_ (10.2 vs. 3.48 mg/L) values systemically below the recommended therapeutic ranges for combined therapy, confirming subtherapeutic CLI concentrations in plasma during CLI-RIF using the oral route. Instead of CLI monotherapy, Bernard et al. administered CLI concomitantly with levofloxacin (500 mg once daily) as control. Lastly, a recent study from Zeller et al. [85] (2021) quantified that during RIF-CLI, median C_min_ was also markedly lower (2.09 vs. 0.18 mg/L), as was median C_max_ (7.95 vs. 1.53 mg/L).

A theoretical target plasma value with regard to C_min_ was defined for the two first discussed studies [79,89], and both studies obtained a C_min_ value in combined treatment below the previously set threshold value. By following the same criteria, target CLI concentrations were similarly set across the studies, and dosage variations for high TBW patients were performed in all studies. The impact of the administration route on the magnitude of the CLI-RIF interaction will be discussed in the following sections.

## 5. CYP3A4-Mediated Drug–Drug Interactions

Among the CYP3A enzyme subfamily, the CYP3A4 isoenzyme is the most abundant in the human liver (~40%) and is implicated in phase I metabolism of more than 50% of all prescribed medications [90,91]. CYP3A4 can recognize and metabolize a wide array of xenobiotic substances, known as CYP3A4 substrates. In addition, induction or inhibition of CYP3A4-metabolized pathways can result in enhanced or suppressed metabolic capacity and drug plasma concentrations, respectively. Because of the key role of CYP3A4 in drug metabolism, such enzyme changes can lead to a PK drug–drug interaction (DDI) while coadministering these CYP3A4 substrates together with CYP3A4-inhibitors or inducers.

CYP3A4 inhibition can result in serious adverse events since the intrinsic CL of the victim drug(s) is reduced, leading to undesirable elevations of plasma drug concentrations [92]. CYP3A4 induction may cause a reduction in the therapeutic efficacy of CYP3A4 substrates, as the victim drug(s) elimination is increased, lowering drug concentrations and provoking a decrease in the victim drug(s) pharmacological effect [93]. Inhibition is an almost immediate response, while induction is a relatively slow regulatory process [92]. Moreover, in the presence of active metabolites, three factors should be considered during DDI interpretation: the metabolic pathway, the ratio of the parent drug (CLI) to its metabolites, and the potency of the metabolites [94]. However, current insight into the potential relevance of CLI’s active metabolites on this DDI is still very limited [92], as quantitative data on their exposure are not yet available. Quantitatively predicting the clinical magnitude of CYP3A-mediated DDIs is complicated, considering that the clinical outcomes depend on contributing factors such as interindividual variability associated with patients and drugs [92,93]. Insufficiency of proper experimental tools also contributes to the difficult prediction of these kinds of DDIs, as the potential of both human interactions is commonly assessed by in vitro models [90,93].

In the following subsections, we will reflect on the clinical relevance of the DDIs that can occur between our victim drugs (CLI and its active metabolites) and specific perpetrators (CYP3A4-inhibitors and inducers) (Table 7), based on the PK and the PD of CLI and its metabolites. Although not covered in this review, the topic of potential PK DDIs could be further extended to food–drug or health products (e.g., Chinese medicine).

### 5.1. CYP3A4-Inhibition: Macrolides and Antiretroviral Drugs—Clindamycin

Some macrolide antibiotics (erythromycin) and antiretroviral drugs (ritonavir) strongly inhibit CYP3A4, potentially leading to an increase in CLI concentration and risk of CLI toxicity or side effects. For macrolides, data suggest that CYP3A4 is subjected to mechanism-based inhibition [90], while the protease inhibitor HIV-1 ritonavir is a competitive and noncompetitive, irreversible CYP3A4-inhibitor [95] (Table 7).

Macrolides are not often combined with CLI in daily clinical practice when used as antibiotics. However, despite being an effective inducer of MLS_B_ resistance, erythromycin is a moderate CYP3A4-inhibitor [96] that might be combined with CLI when administered PO in a low dose (125–250 mg.q12h) as a gastroprokinetic to help control acid reflux. Furthermore, when *Staphylococcus* spp. are tested for CLI susceptibility, inducible MLS_B_ resistance due to erythromycin is tested. Isolates with such inducible resistance are resistant to erythromycin but appear susceptible to CLI in routine in vitro testing. However, clinical failures of CLI therapy for the treatment of MRSA infections have been documented for strains that are CLI sensitive but erythromycin resistant. Therefore, in such a case, CLI would be reported as ‘resistant’.

Regarding antiviral drugs, nirmatrelvir and ritonavir are two coadministered antiviral medications marketed as Paxolavid ^®^ for the treatment of mild to moderate COVID-19 caused by the severe acute respiratory syndrome coronavirus 2 (SARS-CoV-2) virus. Paxolavid ^®^ is administered PO to adults and children (≥12 years old weighing at least 40 kg) in a dose of 300 mg/100 mg (nirmatrelvir/ritonavir) twice a day for 5 days [97]. Paxolavid ^®^ is not an FDA-approved drug, as it is still considered an investigational medicine. The FDA has issued an Emergency Use Authorization (EUA) to make this antiviral available during the COVID-19 pandemic [97]. Nirmatrelvir inhibits proteolysis by binding the 3CL protease, ultimately leading to the cessation of viral replication. Ritonavir is not active against SARS-CoV-2 and acts as a booster agent by potently inhibiting CYP3A4, thereby maximizing the nirmatrelvir concentration in plasma [98]. A patient infected with this virus undergoing CLI treatment may experience a CYP3A4-inhibiting effect due to ritonavir’s role as a CYP3A4-inhibitor. All in all, caution should be given when extrapolating the effect of these CYP3A4-inhibitors (macrolides and antivirals) alone to the effect of combination regimens on drugs with CYP3A4 activities such as CLI.

### 5.2. CYP3A4-Induction: Rifampicin–Clindamycin

Despite many CYP enzymes known to be inducible, CYP3A4 induction is probably one of the most important causes of documented induction-based interactions [99] and a major concern in clinical practice. In contrast to the extensive list of CYP3A4 inhibitors, only a small number of drugs are identified as CYP3A4 inducers. This is the case for rifampicin, a potent CYP3A4-inducing agent that undergoes a transcriptional mode of action since the drug is a specific pregnane xenobiotic receptor (PXR) agonist (Table 7). Moreover, current findings suggest that flucloxacillin might be a CYP3A4 inducer, as flucloxacillin has been found to decrease the levels of the following CYP3A4-metabolized drugs: voriconazole, quinidine, and tacrolimus [100,101,102,103]. For this, recent studies describing the DDIs between flucloxacillin and azole antifungals, e.g., voriconazole and isavuconazole (both metabolized by CYP3A4) [104,105], have been conducted, and it is unclear if this should be extrapolated to other antibiotics such as CLI.

For RIF-CLI interaction, full CYP induction is reached after approximately 1 week, and recovery to baseline activity after rifampicin withdrawal is about 2 weeks [106]. Because direct assessment of CYP induction in vivo through the measurement of enzyme activity is not possible, indirect evaluation can be performed through the comparison of CLI plasma AUC before and after cotreatment with rifampicin. Both administration route and drug’s characteristics need to be considered for this comparison. The route of administration is an important factor since CLI shows a high hepatic CL of ~1470 mL/min considering an average CL value of 0.35 L/h/kg (70 kg standardized TBW). During cotreatment with rifampicin, oral AUC is significantly downregulated, resulting in a greater magnitude of changes in oral AUC compared with increased CL changes. This could primarily be due to an increase in prehepatic first-pass effects but also because hepatic blood flow is much bigger than systemic CL. IV AUC is not sensitive to changes in enzyme activity since systemic CL is not affected by the first-pass effect and is limited by hepatic blood flow. This generalized decrease in systemic exposure is translated into therapeutic and pharmacological efficacy consequences that may lead to dose adjustments. In addition, because of the formation of CLI-active metabolites [5], interpretation of clinical consequences would be somewhat complicated in this case.

Previously mentioned findings from Zeller et al. [85] regarding C_min_ and C_max_ values (Table 6) confirm that the magnitude of RIF-CLI was markedly increased by oral intake, describing a pronounced decrease in concentrations of PO administered CLI. Assuming that CLI plasma AUC before and after cotreatment is likely the most relevant indicator to assess CYP-inductive effect, a ~12-fold decrease comparing monotherapy vs. combined (37.7 vs. 3.1 mg.h/L) of oral AUC_0–8 h_ was reported. In addition, a ~19-fold increase in oral CLI’s CL (7-fold higher compared with the IV route) was observed. According to Zeller et al., this might suggest that rifampicin more specifically increases the prehepatic first-pass effects. These conclusions are not only supported by the variations in the CL but also by the significantly lower bioavailability of oral CLI in monotherapy vs. combined (59% vs. 10%). There was no significant change in T_1/2_ (~1.2 h less in RIF-CLI), independent of the route of administration, supporting the argument that CYP induction has little effect on systemic CL of high hepatic CL drugs such as CLI [93].

## 6. Drug–Disease Interactions: Impact of Inflammation on CYP3A4/5 Activity

Acute and chronic inflammatory responses can be caused by infectious or noninfectious stimuli, leading to the alteration of hepatic functions that are critical for drug metabolism, such as drug-metabolizing CYP isoenzyme activity. Good evidence shows that under disease conditions, hepatic metabolism plays a crucial role in the secretion of inflammatory mediators, which can modulate drug metabolism by reducing CYP expression [107]. One of the key stimulators involved in both acute- and chronic-phase responses is interleukin-6 (IL-6), together with C-reactive protein (CRP), an IL-6-regulated acute-phase protein that is synthesized in the liver when IL-6 levels increase during inflammation. Next to IL-6 and CRP, plasma protein AAG is an essential biomarker to assess inflammation’s clinical impact on CYP-induced drug metabolism. Although long-term investigations on this effect are still scarce, current published data in adults suggest that inflammation has an isoform-specific and intensity-specific impact [108] as a result of pretranscriptional and post-transcriptional mechanisms [107]. CLI being primarily a CYP3A4-substrate means that CYP3A4 drug metabolic rate is reduced, and levels of protein and mRNA subsequently decrease at the level of gene transcription. As a result, there is a decrease in CYP3A4-dependent drug CL and impaired CLI biotransformation during the operation of host defense mechanisms. If an anti-inflammatory drug is administered to treat inflammation disease, mediator levels are lowered, and metabolic capacity appears to return to baseline level when the disease is resolved [108].

A recently published review from C.M. White [109] assessed the concentration of a CYP3A4 substrate with an acute infectious or noninfectious cause of inflammation from a total of 23 studies. Evaluated CYP3A4 substrates in descending order of predominance included quinine (8 studies), midazolam (4 studies), lopinavir (2 studies), erythromycin (2 studies), and other miscellaneous drugs (1 study). A correlation between increasing biomarker concentration and reduced CL of CYP3A4 substrates was found in 10 of 12, 2 of 2, and 2 of 3 studies assessing CRP, AAG, and IL-6, respectively. This association is especially strong for benzodiazepines. Several studies showed large changes in drug concentration/dose ratio, AUC, or CL, in patients with inflammation. However, quetiapine had a small increase in the concentration of only 11% among people with CRP concentrations > 5 mg/L, which is qualitatively less robust than what was seen with benzodiazepines (midazolam, alprazolam), perampanel, and antimicrobial agents (erythromycin, cyclosporin). In assessed studies concerning populations with malaria [110,111,112,113,114,115], when patients developed an acute malarial infection, CYP3A4 substrate drug concentrations were elevated or CL was reduced, even though no markers of inflammation were concomitantly assessed. Moreover, patients with acute malarial infection had a much higher AUC_0–12h_ compared with convalescent patients (37.9 ± 14.7 vs. 17.9 ± 8.5 mcg/mL) [115].

Given that inflammation is an important determinant contributing to variation in CYP activities between and within individuals [107], an effort must be made to better understand the impact of clinically useful inflammatory biomarkers released during inflammation, such as IL-6, CRP, and AAG. These mediators can measure the severity of inflammation, that being proportional to the potential suppression of CYP3A4 activity, and evaluate the clinical effectiveness of CYP3A4 substrates in reference and special populations. Mechanistically, understanding enzyme specificity and mechanisms of regulation will allow us to improve drug efficacy or safety, improve knowledge of acute and chronic diseases, and personalize patients’ drug regimens. Moreover, transcriptional effects have a great impact on CYP3A4 substrates with a relatively narrow therapeutic index such as CLI since an increase in toxicity can more easily occur and complicate therapeutics-causing adverse events (AEs). For this, clinicians with patients on narrow therapeutic index CYP3A4 substrates must monitor their patients more judiciously when new infections or other inflammatory stimuli occur to prevent AEs or loss of drug efficacy [109].

## 7. Safety and Adverse Event Profile

Even if CLI is generally a well-tolerated drug, the SmPC describes some AEs that can be resolved with dose adjustment or discontinuation of the antibiotic [51]. CLI’s most frequent AEs with systemic administration are mainly GI, resulting from CLI substantial changes in the GI tract’s healthy flora [51]. The destruction of much of this microflora may result in the development of *Clostridium difficile*-associated diarrhea (CDAD). Diarrhea is the most common AE and occurs in up to 20% of patients, and CDAD may occur more frequently compared with other oral agents [116]. Other common GI AEs include nausea, vomiting, or stomach pains. A more severe and also uncommon GI that may develop (during or after treatment) is *C. difficile*-associated pseudomembranous enterocolitis (PMC). This potentially life-threatening infection is often associated with CLI and is caused by a complication of CDAD, which can lead to fulminant colitis, sepsis, and death. Growing PMC incidence corresponds with the increased use of wide-spectrum antibiotics in hospitalized patients [117]. Although PMC may affect all age groups, the incidence is low in the pediatric SPP. Mortality is rare in pediatrics and involves patients with serious coexistent illness, infection, or congenital disabilities [118]. Regarding pregnant women, an association between *C. difficile* infection and pregnancy has not been stressed [119]. The occurrence of PMC for both discussed SPPs (pediatrics and pregnant, breastfeeding, and postpartum women) is of particular concern, and further research into the scope and risk factors for children and peripartum PMC is warranted.

A recently published systematic review [120] updating the evidence for associations between antibiotic classes and *C. difficile* infection stated the modest association observed between clindamycin and such disease. This up-to-date synthesis of evidence in relation to this potential risk suggests that CLI is relatively safe or not the most unsafe drug compared with other antibiotics (e.g., carbapenems, third- and fourth-generation cephalosporins). Nonetheless, this relevant complication should be considered during CLI’s use. Besides, because predicting risks for DDIs involving CYP3A4-induction and inhibition is difficult, caution and TDM are needed when administering CLI together with CYP3A4-substrates, and dose adjustment of these substrates might be necessary. Moreover, CYP3A4 metabolism-mediated PD (side) effects should also be considered and assessed.

## 8. Discussion

In this review, we explored the pharmacology and PKs of CLI in a reference population and selected SPPs (pediatrics, pregnant, breastfeeding, and postpartum women). Our conclusion is that there is a major knowledge gap regarding the potential PD and PK relevance of CLI’s active metabolites in daily clinical practice.

Important observations on the PK/PD relevance of CLI’s activity metabolites were already reported by Wynalda et al. [5] in 2003, as this group examined the in vitro metabolism of CLI in human liver and intestinal microsomes. Following incubation with human liver, ileum, and jejunum microsomes and recombinant CYP3A4 and CYP3A5 microsomes, CLI sulfoxide was found to be the major metabolite formed, accounting for more than 90% of total consumed CLI. A minor metabolite, N-demethyl CLI, was also identified. Only CYP3A4 and, to a certain extent, CYP3A5 were able to catalyze the formation of CLI sulfoxide, and only inhibitors of CYP3A could attenuate the rate of CLI’s sulfoxide formation. In addition, in vitro findings showed that CLI did not inhibit the metabolic activity of other CYPs (e.g., CYP1A2, CYP2C9, CYP2C19, CYP2E1, or CYP2D6) and only moderately inhibited CYP3A4. These data provide evidence of the mechanism of the DDI between CLI and the aforementioned CYP3A4 inducers. Because of the limited data on the involved biotransformation routes (Figure 2, Table 1), determining to what extent the altered CLI metabolite exposure results in altered CLI’s drug efficacy or safety is not yet possible. Furthermore, metabolites quantification in plasma and other human matrices (e.g., urine, bile, bone, amniotic fluid, etc.) is still to be determined and must be studied in the future to evaluate their potential relevance to CLI’s disposition.

Regarding CLI’s clinical pharmacology and therapeutics for the selected SPPs, future research is urgently required, especially regarding topics such as dose optimization and adapted drug therapy. Based on limited data on PK and efficacy and safety in the SPP background, we claim that further studies should be designed and carried out to facilitate the efficient evaluation of CLI’s PKs. Moreover, experimental CLI-RIF administration for the treatment of BJIs is based on performed PK studies (Table 6). A key limiting factor of the discussed studies is the low number of participants for most of the studies, leading to a low power statistical analysis. Furthermore, a detailed methodological description was even absent in the report by Bernard et al. [89]. Overall, the bioavailability of CLI decreases significantly in the setting of induction by rifampicin during the CLI-RIF cotreatment. Given the marked increase, through oral intake, in the magnitude of RIF-CLI, optimal CLI dose in combination with rifampicin remains to be determined, and close-target drug monitoring (TDM) of drug and bioactive metabolites is advised to guide dosage adjustments. Moreover, all previous investigations only considered the CLI, and, to our knowledge, the potential relevance of CLI’s metabolites in the PK/PD target remains unexplored. In the future, the performance of PK studies, including CLI’s metabolites in a larger number of both healthy and ill patients, will reflect best clinical practice regarding the CLI-RIF CYP3A4-inducing DDI.

Additionally, because SPPs possess CYP3A4 altered activity, clinical implications of CYP3A4 induction and inhibition must be evaluated for these special populations and compared to healthy adult populations in the future. Overall, awareness of these DDIs will contribute to safer and more controlled management of patients receiving treatment for the mentioned systemic infections. Given the DDI metabolic variation involved, caution and close TDM are needed when CLI and CYP3A inducers and inhibitors are simultaneously administered. Dose optimization might be necessary, certainly for antimicrobials such as CLI, likely to undergo therapeutic failure because of underdosing and low exposure. Regarding the impact of inflammation on CYP3A4/5 activity, many drugs were assessed in C.M. White’s review [109]. Nevertheless, the extent to which the results from CYP3A4-metabolized drugs can be applied to all CYP3A4 substrates remains unclear. To our knowledge, no study has directly evaluated inflammation-induced CYP expression in standard or other populations for CLI.

Additional studies will be needed to obtain more data on CLI’s PK and PD, including the metabolites, in order to further improve the assessment of CLI’s efficacy and safety. Moreover, efforts should be made to discover novel strategies in the approach to the treatment of bacterial infections in special patient populations.

## 9. Conclusions

Based on the paucity of data found on CLI’s PKs, efficacy and safety in the field of systemic infections, we claim that efforts should be made to discover new strategies in the approach to antimicrobial infection with respect to the role of CLI, to facilitate the evaluation of antimicrobial prophylaxis and treatment in this specific setting. To date, knowledge of CLI’s active metabolite contribution to the efficacy and safety of CLI is scarce and inconclusive. In conclusion, identification of the current knowledge gaps (SPPs, DDIs) with respect to the pharmacology and PKs of CLI is crucial to developing a research strategy toward precision medicine.

## Figures and Tables

**Figure 1 antibiotics-11-00701-f001:**
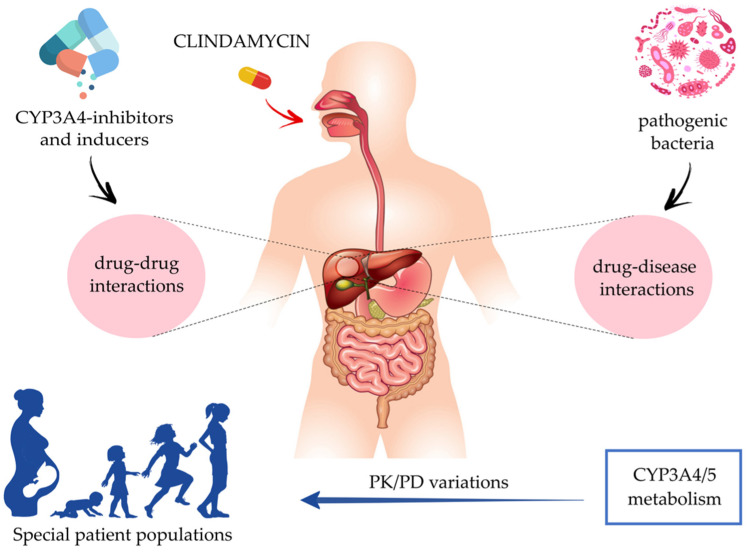
Schematic overview of the main topics addressed in this review.

**Figure 2 antibiotics-11-00701-f002:**
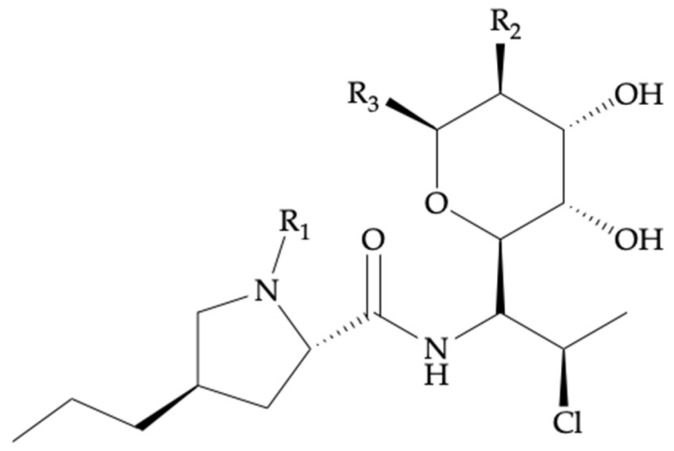
Chemical structure for clindamycin and related components.

**Table 1 antibiotics-11-00701-t001:** Structural information for clindamycin and related components.

Compound Name	Biotransformation	R_1_	R_2_	R_3_
Clindamycin	-	-CH_3_	-OH	-SCH_3_
Clindamycin phosphate	-	-CH_3_	-OPO_3_H_2_	-SCH_3_
Clindamycin palmitate	-	-CH_3_	-OCOC_15_H_31_	-SCH_3_
Clindamycin sulfoxide ^1^	S-Oxidation	-H	-OH	-SCH_3_
N-demethyl clindamycin ^1^	N-Dealkylation	-CH_3_	-OH	-SOCH_3_

^1^ These compounds are bioactive metabolites of clindamycin, undergoing human phase I metabolic reactions.

**Table 2 antibiotics-11-00701-t002:** Summary Table of clindamycin therapeutics.

Clindamycin	
Chemical nomenclature	7-chloro-7-deoxy-lincomycin
Chemical structure	See Figure 2, Table 1
Pharmacotherapeutic group	Lincosamides
Indications ^1^	1. Surgical prophylaxis in the event of 2. Beta-lactam allergy; 3. Prophylaxis and treatment of pregnancy infections; 4. Treatment of diabetic foot infections; 5. Treatment of bone and joint, fracture-related, and periprosthetic joint infections
Mode of action	Bacterial protein synthesis inhibitor. Binds to 50S ribosome and inhibits peptidyl transferase and translocation
Route of administration	PO ^2^ (CLI HCl, palmitate HCl); IV ^2^ (CLI phosphate)
Formulations	PO (capsules, solution); IV (injection solution)

^1^ Concerning prophylaxis and treatment of systemic bacterial infections; ^2^ PO—oral administration; IV—intravenous administration.

**Table 3 antibiotics-11-00701-t003:** Clinical breakpoints for clindamycin’s most relevant bacterial pathogen strains.

Bacteria Type	Pathogen	Clinical Breakpoints ^1^
Gram-positive aerobes	*Staphylococcus* spp.	0.25
	*Streptococcus* spp. ^2^	0.5
	*Scheme 0.*	0.5
Anaerobes	*Fusobacterium necrophorum*	0.25
	*Prevotella* spp.	0.25
	*Bacteroides* spp.	4 ^3^

^1^ The concentrations are given in mg/L; ^2^
*Streptococcus* spp.: *Streptococcus* groups A, B, C, and G.

**Table 5 antibiotics-11-00701-t005:** Clinical practice and efficacy indications for clindamycin, concerning prophylaxis and treatment of systemic bacterial infections.

Indications	Predominant Causative Pathogens ^1^	Type of Treatment	Admin. Route	Adult Dosing	Followed Guidelines
Surgical prophylaxis (SSIs ^2^) in the event of beta-lactam allergy	Clean procedures: *S. aureus*, CoNS ^2^ Clean-contaminated procedures: GN ^2^ spp., *Enterococcus* spp. ^3^	Second line. Monotherapy or combined treatment	IV ^2^	600 mg.q6h (<70 kg) 900 mg.q6h (≥70 kg)	ASHP ^2^, IDSA ^2^, SIS ^2^, SHEA ^2^,
Prophylaxis and treatment of pregnancy infections	*S. aureus*, CoNS, group B *Streptococcus*	Second line. Maternal allergy to penicillins	IV	900 mg.q8h until delivery ^4^	CDC ^2^, American College of Obstetricians and Gynecologists
Treatment of DFIs ^2^	*S. aureus*, beta-hemolytic streptococci, GN spp.	Second line. Severe beta-lactam allergy. Combined treatment ^5^ in the case of IV	Mild DFI: PO ^2,5^. Moderate or severe DFI: IV ^5^	PO: 300–450 mg.q8h IV: 600 mg.q8h	IDSA
Treatment of BJIs ^2^, FRIs ^2,^ and PJIs ^2^	*Staphylococcus* spp., CoNS, *Streptococcus* spp., *Enterococcus* spp., *Pseudomonas aeruginosa*, anaerobic bacteria	Combined treatment with rifampicin. See Section 4.4 T	~6 weeks IV + ~12 weeks PO ^6^	IV or PO: 600 mg.q8h	Consensus from an International Expert Group [52]

^1^ Displayed in order of prevalence; ^2^ SSIs—surgical site infections; DFIs—diabetic foot infections; BJIs—bone and joint infections; FRIs—fracture-related infections; PJIs—periprosthetic joint infections; CoNS—coagulase-negative staphylococci; GN—Gram-negative; IV—intravenous administration; PO—oral administration; ASHP—American Society of Health-System Pharmacists guideline on Antimicrobial Prophylaxis in Surgery; IDSA—Infectious Diseases Society of America guidelines; SIS—Surgical Infection Society guidelines; SHEA—Society for Healthcare Epidemiology of America; CDC—Centre for Disease Control; ^3^ Clindamycin is not active against *Enterococcus* spp. ^4^ In the case of prophylaxis for group B *Streptococcus* neonatal disease; ^5^ In the case of mild DFIs PO route is recommended, while moderate to severe coinfections can be temporarily treated by IV coadministration of clindamycin with ciprofloxacin or levofloxacin. ^6^ PO administration for up to 12 weeks applies in case an implant is present.

**Table 6 antibiotics-11-00701-t006:** Summary of relevant prospective pharmacokinetic studies for cotreatment of clindamycin and rifampicin in bone and joint infections.

Pharmacokinetic Studies *	Posology and Route of Administration		Theoretical Target Plasma Concentration ^1^	Measured Plasma Concentration ^1^	Measurement Technique ^1^
[Reference]	CLI ^4^	RIF ^4^		Monotherapy vs. Combined	
Curis et al. [79]	600 mg.q8h, PO/IV ^4^ bolus	NS, PO/IV bolus	C_min_ ^4^ = 1.7	C_min_ ^2,4^ = 1.36 vs. 0.29 C_max_ ^2,4^ = 7.48 vs. 4.46	HPLC-UV ^4^
Bernard et al. [89]	600 mg.q8h, PO ^4^	600 mg.q12h, PO	C_min_ = [2,3,4] C_max_ ^4^ = [5,6,7,8]	C_min_ ^3,4^ = 4.7 vs. 0.79 C_max_ ^3,4^ = 10.2 vs. 3.48	NS ^4^
Zeller et al. [85]	2400 mg/day, IV ^4^ infusion; 750 mg.q8h, PO	600 mg.q12h, PO	C_ss_ ^4^ = [5,6,7,8]	C_min_ ^2,4^ = 2.09 vs. 0.18 C_max_ ^2,4^ = 7.95 vs. 1.53	LC-MS/MS ^4^

^1^ Regarding clindamycin in monotherapy vs. combined with rifampicin. The concentrations are given in mg/L; ^2^ Median values; ^3^ Mean values; ^4^ RIF—rifampicin; CLI—clindamycin; NS—not specified; C_ss_—target steady-state concentration; C_min_—trough concentration; C_max_—peak concentration; LC-MS/MS—liquid chromatography coupled with tandem mass spectrometry; HPLC-UV—high-performance liquid chromatography with UV detector; IV—intravenous administration; PO—oral administration; PO/IV—administration of both routes; * Dosage variations for high total body weight (TBW).

**Table 7 antibiotics-11-00701-t007:** CYP3A4-mediated drug–drug interactions arising from combined therapies with clindamycin.

Type of CYP3A4- Mediated DDI ^1^	Drug	Drug Class	DDI ^2^ Mechanism	DDI ^2^ Potency	Indication	Type of Combined Treatment ^3^	Admin. Route	Adult Dosing of Drug
**CYP3A4**- **inhibition**	Erythromycin	Macrolide antibiotic	Mechanism-based inhibition	Moderate inhibition	Gastroprokinetic: control acid reflux	Combined in low doses	PO ^1^	125–250 mq.q12h
	Ritonavir	Antiretroviral: protease inhibitor HIV-1	Competitive and noncompetitive, irreversible inhibition	Potent inhibition	Mild to moderate COVID-19 ^1^ caused by the severe SARS-CoV-2 ^1^ virus	Paxolavid ^®^ (nirmaltrevir/ ritonavir)	PO	Paxolavid ^®^ (300 mg/100 mg).q12h for 5 days
**CYP3A4**- **induction**	Rifampicin	Rifamycin antibiotic	Transcriptional PXR ^1^ agonism	Potent inhibition	Treatment of BJIs ^1^	See Table 6	PO or IV ^1^	See Table 6

^1^ DDI—drug–drug interaction; PO—oral administration; IV—intravenous administration; COVID-19—coronavirus disease 2019; SARS-CoV-2—severe acute respiratory syndrome coronavirus 2; PXR—pregnane xenobiotic receptor; BJIs—bone and joint infections; ^2^ Regarding CYP3A4 inhibition or induction; ^3^ Combined treatment with clindamycin.

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
