# Peer review of "Ways to Improve Insights into Clindamycin Pharmacology and Pharmacokinetics Tailored to Practice"

_antibiotics, 2022, doi:10.3390/antibiotics11050701_

Round 1

Reviewer 1 Report

Laura Armengol Álvarez et al described the review manuscript 'Ways to Improve Insights into Clindamycin Pharmacology and Pharmacokinetics Tailored to Practice'. This review comprises the pharmacology and pharmacokinetics of Clindamycin in special populations, drug-drug interactions, drug-disease interactions, clinical practice and efficacy and safety and adverse event profiles. 

This manuscript lacks the figure or table summarizing the content of the manuscript for easy understanding to the scientific community. 

It would be useful for readers if the authors provide tables or figures related to each section of the manuscript. (special populations, drug-drug interactions, drug-disease interactions, clinical practice and efficacy and safety and adverse event profiles).

Author Response

Thank you for your useful comments and suggestions. Upon them, a summary figure (Figure 1) has been designed and added for easier understanding. Moreover, three tables (Table 4, 5 and 7) for ‘special populations’, ‘drug-drug interactions’ and ‘clinical efficacy’ sections have been inserted, as considered convenient for the readership.

Reviewer 2 Report

  1. I recommend specifying the type of the review in the title, including the term narrative review or overview or update
  2. I recommend reorganizing the review in logical sections. Introduction is too long. Explain the rationale for the review in the context of what is already known and provide an explicit statement of questions being addressed. The aim of the review should be defined in the introduction. The pharmacological and pharmacokinetic data could represent a separate section.
  3. Lines 551-554: “In the following subsections we will assess the magnitude of changes in PK/PD parameters that occur during DDI between our victim drugs (CLI and its active metabolites) and specific potent perpetrators (CYP3A4-inhibitors and inducers) to determine if the expected or observed interactions are clinically significant.” Unfortunately, preclinical, and clinical data concerning DDI between CLI and CYP3A4-inhibitors or flucloxacillin are lacking, so the magnitude of changes in PK/PD parameters cannot be assessed. It is only possible to say that DDI are possible.
  4. Lines 681-704: The major disadvantage of clindamycin is its propensity to cause antibiotic-associated diarrhea, including Clostridium difficile colitis. Unfortunately, this aspect is superficially addressed. Make sure you present a critical discussion, not just a short descriptive summary of the topic.
  5. The discussion section suggests the need for additional studies for clindamycin, but this section requires further work to clarify what is recommended to the reader.
  6. Please check reference style.

Author Response

Thank you for your useful comments and suggestions. Please find in the attachment a PDF with the answers.

Reviewer 3 Report

Suggestions: An addition of a table illustrating the Pharmacokinetics (PK) in Special patient populations (SPP) for Section 1.3. maybe great in addressing the various challenging issues given the number of variants and unknowns. The table helps a lot in summarizing the lengthy description. This may also serve as a ground format for future expansion/additions to include cancer and COVID-19 patients.

Aside from that, it is applicable and useful for the research community where Chinese medicine is used in integration with western medicine, especially in the area of PK drug-drug interaction (DDI).

Author Response

Thank you for the useful comments and suggestions. Upon them, three tables (Table 4, 5 and 7) for ‘special populations’, ‘drug-drug interactions’ and ‘clinical efficacy’ sections have been inserted, as considered convenient for the readership. Moreover, a comment on the consideration for other special populations like cancer or COVID-19 patients has been inserted in the section regarding ‘special patient populations’. Some lines on the further extension/applicability of the drug-drug interactions discussed to food-drug or health products (like e.g., Chinese medicine) have also been included.

Round 2

Reviewer 2 Report

Thank you for responding to the comments.